# First Insights into the Home Range of an Adult Male Mediterranean Monk Seal *Monachus monachus* in the Ionian Sea, Greece, and Considerations About the Future Management of the Area

**DOI:** 10.3390/ani15050617

**Published:** 2025-02-20

**Authors:** Aliki Panou, Spyros Tsoukalas, Anastasios Anestis, Luigi Bundone

**Affiliations:** 1Archipelagos—environment and development, 28100 Kefalonia, Greece; 2Department of Philosophy and Cultural Heritage, Ca’ Foscari University of Venice, Dorsoduro 3484/D, 30123 Venezia, Italy; 3Archipelagos—ambiente e sviluppo Italia, 30132 Venezia, Italy

**Keywords:** marine mammal, Monachinae, use of caves, establishment of MPAs, central Mediterranean, Greece

## Abstract

An adult male Mediterranean monk seal repeatedly used marine caves for resting on two islands in the Greek Ionian Sea, about 15 km apart—Kefalonia and Zakynthos. This distance can be considered as the minimum distance within the home range of the animal that needs to be taken into account within the framework of the ongoing elaboration of conservation measures.

## 1. Introduction

The Mediterranean monk seal (*Monachus monachus*, Hermann 1779), referred to hereafter as the MMS, is a highly threatened marine mammal with an estimated global population of less than 1000 animals, half of which live in the Mediterranean Basin [1]. Its ecology remains largely unknown, while understanding the species’ use of both marine and terrestrial habitats, as well as its patterns of movement and home range, is crucial for designing effective monitoring programmes and for formulating adequate conservation measures.

Historical records indicate that the MMS used to haul out on open beaches and aggregate in colonies consisting of many individuals, as is the case in other pinniped species. Currently, MMS individuals rest and give birth mainly in secluded marine caves with internal beaches [2]. The social structure typical of a seal colony has in fact been retained only in the Atlantic sub-population in Cabo Blanco, Mauritania/Western Sahara. All other nuclei are characterized by aggregations of a limited number of individuals [1].

Studies on habitat use have revealed complex interactions with environmental factors, explaining patterns of species distribution and abundance [3]. In this context, home range—defined by Burt (1943) [4] as the “area traversed by the individual in its normal activities of food gathering, mating, and caring for young”—is an important parameter for the distribution of a species.

Several studies have documented the MMS’s remarkable ability to undertake long-distance travels, swimming along coastlines/islands or through open sea covering areas of up to 300 km [5,6]. Hence, movements along coasts and between islands are generally to be expected, but their frequency and extent are barely known. In the Mediterranean Sea, only one comprehensive study has been carried out so far on the home range of this species, estimating the coastal habitat partitioning for adult male MMSs to range between 37 and 56 km [7].

In the Ionian region, a viable monk seal sub-population exists only in the Greek waters of the central and southern Ionian Sea. This sub-population is genetically slightly different from that in the Levantine and the Aegean Seas and also of that formerly present in the western Mediterranean Sea [8,9,10]. The MMS in the central Ionian Sea has systematically been studied over the period from 1985 to 2002. Among other issues, marine caves with an internal beach were regularly surveyed to document their use by seals for resting, moulting, and reproduction [11,12,13]. The same method was applied on Zakynthos Island, southern Ionian Sea, for shorter periods of time, i.e., from 1990 to 1992 and from 1997 to 1999 [14,15,16].

Since 2018, a long-term systematic monitoring of 15 caves, using IR cameras, has been carried out in the central Ionian Sea (islands of Kefalonia and the islets east of Ithaca), aiming to photographically identify individual seals, with very encouraging results [17,18]. Based on the data analyzed up until now, 28 different adult and subadult individuals have been identified: 10 males, 13 females, and 5 seals of yet-unknown gender (Archipelagos) [19]. Additionally, a large network of observers in the wider area, including Zakynthos Island, are contributing valuable information on seal sightings, along with the provision of pictures and videos.

Within the above framework, one of the main aims of our long-term study is to determine actual seal numbers and the structure of the seal population and its dynamics in the central Ionian Sea. In parallel, the use of terrestrial habitats is being investigated in order to prioritize marine caves of particular importance such as pupping and/or resting caves. Both aspects are crucial for establishing effective conservation measures. Here, we present a case study of an adult male monk seal that has been registered in resting sites in the southern part of Kefalonia and in the northwestern part of Zakynthos islands in the central and southern Ionian Sea respectively. Furthermore, we draw conclusions regarding necessary conservation and management measures, particularly with respect to the presently ongoing elaboration of protection measures for the entire central and southern Ionian Sea.

## 2. Materials and Methods

The present study focuses on a section of the Greek Ionian Sea extending from southern Kefalonia to northwestern Zakynthos, located in the central and southern Ionian Sea respectively (Figure 1). The sites are located in two different marine NATURA 2000 sites (and future MPAs): GR2220007 in Kefalonia and GR2210001 in Zakynthos.

The two areas are similar in terms of oceanographic and climatic conditions, as well as with respect to food availability for the seals and possible competitors. Furthermore, they are both located in uninhabited regions where only fishing takes places throughout the year. During the summer months, boat trips along these coastlines are carried out for tourists.

Our cameras are installed in up to 15 marine caves throughout the central Ionian Sea, which is known from previous studies to have been used by seals [11]. Zakynthos Island is not part of the area covered by our photo identification project. Nevertheless, a network of observers, established in previous projects, still contribute by providing data on sightings.

Several camera models, adapted to better resist the hostile environment (e.g., humidity, variation of pressure, and salt sprays), have been used since the beginning of the monitoring activities, e.g., UoVision UV595 HD (Uovision Europe, Kankaistentie 4, 51200 Kangasniemi, Finland), KeepGuard KGH695 & Guard1 (HuaRui Technology (ShenZhen) Co., Ltd., Shenzhen City, Guangdong Province, China), and Reconyx Hyperfire 2 (Reconyx Inc., 3828 Creekside Ln, Ste 2, Holmen, WI, USA). In the cave in southern Kefalonia, the camera is installed at a distance of ~7 m from the main part of the beach where seals usually haul out.

Photographic material and videos obtained from the cameras, as well as from opportunistic surveys and citizen science, are continuously being analyzed using a photo identification method with strict criteria in order to avoid an over-estimation of seal numbers, as described in Bundone and Panou (2023) [20]. Seal identification is based on each seal’s natural marks, i.e., patterns of fur coloration and the presence of scars, along with their gender (if detectable).

## 3. Results

Within the framework of the work carried out in the central Ionian Sea, an adult male using a cave in southern Kefalonia, i.e., cave KEF64, was recently identified by its specific features as seal A028 (or “SPYROS”, Figure 2).

By examining the material from the camera, the presence of A028 inside cave KEF64 was repeatedly recorded over the period from 2021 to 2023. In some cases, other seals were present in the cave together with seal A028 (see Table 1). As in all other cases of animals that have already been identified, a thorough review of all previous material is necessary to define in detail the potential use of this cave by seal A028 for the years before 2021, as well as the potential use of other caves in the central Ionian Sea.

By reviewing the material received from the coast of northwestern Zakynthos through our local network of observers (citizen science contribution) and comparing it with the identified adult male individuals from the central Ionian Sea, seal A028 was recognized in a series of pictures. A028 was photographed in the first weeks of September 2023 on two different days in two different coves of a series of caves and overhangs in a bay of northwestern Zakynthos adjacent to each other. Unfortunately, the exact dates of these two sightings were not noted by the observer. Photographs of A028 were also taken at this location on 11 May 2024 (see Table 1 and Figure 3).

After careful examination of the pictures taken in northwestern Zakynthos, the cave (ZAK-B) of the sighting in 2024 was identified as the same cave where seal A028 was also resting in 2023 (Figure 3, pictures centre and right). Cave ZAK-A (Figure 3, picture left) was identified as a cave located next to cave ZAK-B (see Table 1).

Despite the fact that the material obtained lacks high resolution, the main morphological characteristics (i.e., belly patch and scars) were clear enough to allow for the identification of this individual as seal A028. Figure 4 and Figure 5 highlight the most evident features referring to the belly patch of seal A028 from the pictures obtained by cameras in southern Kefalonia and via citizen science in northwestern Zakynthos.

The distance between the cave in southern Kefalonia and the bay with the series of caves and overhangs in northwestern Zakynthos, which includes the two caves where A028 was recorded, is 15 km.

## 4. Discussion

Our results confirm what has long been suspected—adult male seals, at least, do use marine caves on the coasts of both the islands of Kefalonia and Zakynthos, which face each other, by covering a distance of no less than 15 km over open sea. In previous studies, [11] as well as during the present long-term photo identification project launched in 2018 [21], movements from cave to cave covering up to 80 km and some shifts in cave preferences have been documented for the central Ionian Sea. Seal travels from/to Zakynthos, however, have hitherto never been recorded.

In autumn 2023 and again in spring 2024, seal A028 was repeatedly photographed in a series of caves and overhangs in northwestern Zakynthos. Thus, the use of these caves has not been a single opportunistic visit but an active choice of this animal in at least two different seasons half a year apart. Altogether, and within one year, seal A028 was in Kefalonia at least twice (April 2023 and November–December 2023) and was also in Zakynthos at least twice (September 2023 and May 2024). So far, it is not known where seal A028 was during the in-between periods.

Due to the opportunistic collection of data from northwestern Zakynthos, as opposed to the permanent systematic monitoring in southern Kefalonia, it is not known whether this specific seal is actually using more caves in Zakynthos than those recorded in our pictures. This is, however, quite probable since it also used the cave in southern Kefalonia. For the same reason, it is also unknown whether this seal is mostly using caves in Zakynthos and occasionally visits caves in Kefalonia or vice versa, or whether it uses both areas equally along with other caves, unexplored to date. Here, only a parallel systematic photo identification project covering both coastlines would shed light on the use of caves of this (and other) monk seal(s), as well as on the frequency of such movements. The fact that another adult male, the monk seal under code A017, that has frequently been using the cave in southern Kefalonia for six consecutive years has never been registered in any other cave in the central Ionian Sea [22] probably indicates that this animal might prefer to use some cave(s) in Zakynthos and/or elsewhere rather than the caves in the rest of the study area in the central Ionian Sea. Here, again, only a parallel photo identification project covering the coastlines of both islands can provide useful insights.

From the overall evidence to date, not only do adult males move along the above coastlines, though their range is still unknown, but so do seals of other genders and developmental stages. Seal sightings from our network of citizen science were reported in the mid-1990s from the middle of the channel between Kefalonia and Zakynthos (Archipelagos) [23], in which a very young animal was present in 2018 as well (Panos Dendrinos/MOm, personal communication) [24]. In the present case, adult male A028 definitely used caves 15 km apart located on two different islands. This distance marks this seal’s minimum distance of its home range, in accordance with the statement by Burt (1943) [4]. It is unknown whether the animal also uses caves located further apart.

The repetition in the use of the two coasts over time and the short distances between them should disprove that this behaviour reflects “occasional sallies outside the area, perhaps exploratory in nature”, [that] “should not considered part of the home range”, according to Burt (1943) [4]. According to the same author’s definition, “homing”—as also described for several terrestrial and marine mammals, e.g., Ostfeld and Manson (1996), Born et al. (2005), Matsuura et al. (2011) [25,26,27]—can be excluded here as well.

Data from the central Ionian Sea over the last 8–10 years have proven the progressive use of new caves that have almost never been used in the ~30 years before, as well as the existence of reproduction in caves where births have never been recorded before [17]. This general tendency to use an ever-increasing number of caves may drive the animals to more frequent movements between Kefalonia and Zakynthos in future, in search of new terrestrial habitats. Here, social learning and animal culture, as described by Brakes et al. (2021) [28], might be of particular importance within the context of future measures in conservation management.

It is a well-known fact that, on a global scale, home ranges of highly mobile marine megafauna often do not match MPA boundaries [29]. The two marine NATURA 2000 sites in southern Kefalonia and northwestern Zakynthos, respectively, are such examples—they are both future MPAs, independent of each other. Any future management plan for an MPA in either of these sites would not take into account the home range of the MMS as described here. However, the currently ongoing study for the establishment of a future national marine park in the Ionian Sea, fully covering its entire central and southern parts and far beyond towards the south (i.e., the southern tips of the Peloponnese and the islands of Kythira and Antikythira), is an excellent opportunity for addressing this problem.

As of 31 December 2024, an MPA was established by Ministerial Decision around the two minuscule islets of Formicula in the central Ionian Sea, barely half a square kilometre in size (Government Gazette D 13267, 31 December 2024), without taking into account a seal’s home range and use of habitat in the wider area. Within the framework of the above-mentioned study, appropriate conservation measures need to take into consideration the fact that seals do move between Kefalonia and Zakynthos (and most probably also other Ionian islands) and that the entire area has to be considered as a single “conservation unit” rather than elaborating protection measures on a small scale around a couple of resting and pupping caves alone. The need to take into account broader geographical scales when designing MPAs for the MMS has also been recommended by Dendrinos et al. (2007) [6].

## 5. Conclusions

Within the framework of the ongoing study for the establishment of a future national marine park in the Ionian Sea mentioned above, it is necessary to take into account the results presented here, including the species’ home range, for substantially covering the MMS’s critical terrestrial habitats throughout the future Ionian national marine park, thus safeguarding essential marine caves for resting and reproduction. A well-designed full network of marine caves under strict protection, each surrounded by a buffer zone, is needed here, including the so far inadequately studied island of Zakynthos.

## Figures and Tables

**Figure 1 animals-15-00617-f001:**
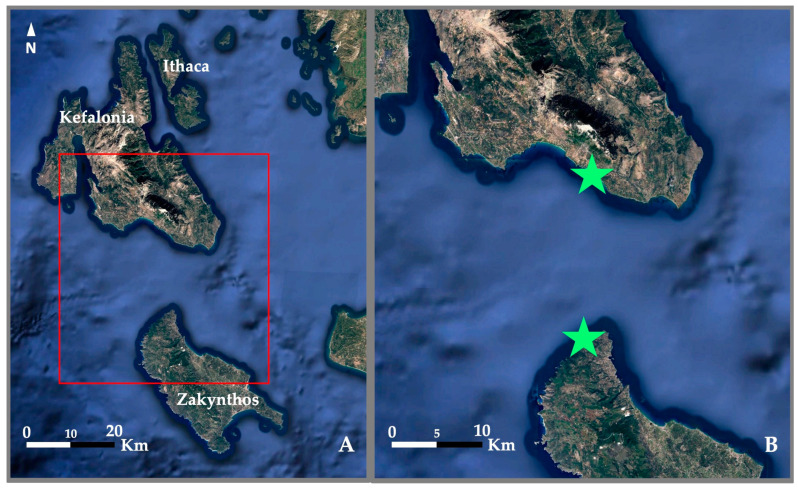
(**A**) Map of Kefalonia, Ithaca, Zakynthos, and minor islets, Ionian Sea, Greece. (**B**) Close-up of the area containing the caves used by seal A028 in southern Kefalonia and northwestern Zakynthos (red rectangle in Figure 1A). The caves’ locations are indicated with green stars. The map was adapted from Google Earth Pro (Google LLC, Mountain View, CA, USA).

**Figure 2 animals-15-00617-f002:**
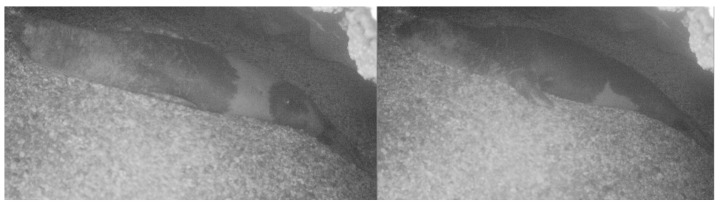
Seal A028 in cave KEF64, southern Kefalonia, 17 February 2023 (camera pictures).

**Figure 3 animals-15-00617-f003:**
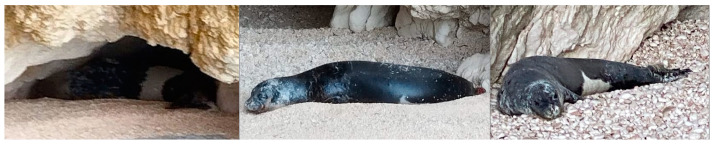
Seal A028 in caves ZAK-A and ZAK-B (pictures left and centre), northwestern Zakynthos, mid-September 2023, and in cave ZAK-B on the 11 May 2024 (picture on the right).

**Figure 4 animals-15-00617-f004:**
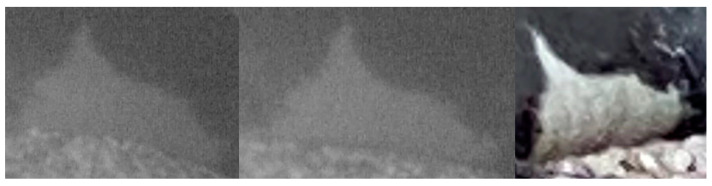
Seal A028, left lateral part of the belly patch: camera pictures from Kefalonia (the first two pictures from the left) and picture from Zakynthos (picture on the right).

**Figure 5 animals-15-00617-f005:**
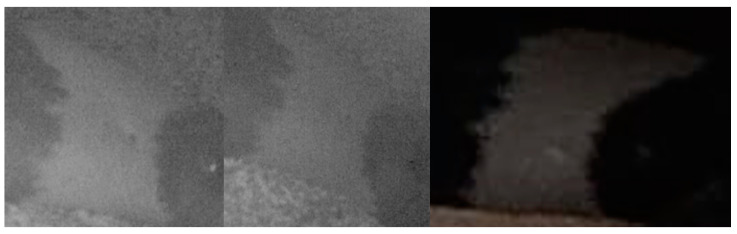
Seal A028, ventral view of the belly patch: camera pictures from Kefalonia (the first pictures from the left) and picture from Zakynthos (picture on the right).

**Table 1 animals-15-00617-t001:** Documented presence of seal A028 in southern Kefalonia and in northwestern Zakynthos.

Date	Notes	Locations	Cave
1 August 2021	With 2 more seals	southern Kefalonia	Cave KEF64
4–5 August 2021		southern Kefalonia	Cave KEF64
27 January 2022	With 2 more seals	southern Kefalonia	Cave KEF64
13–15 February 2023		southern Kefalonia	Cave KEF64
16–17 February 2023		southern Kefalonia	Cave KEF64
16 April 2023		southern Kefalonia	Cave KEF64
Ca. mid-September 2023		northwestern Zakynthos	Cave ZAK-A
Ca. mid-September 2023		northwestern Zakynthos	Cave ZAK-B
3 November 2023	With 1 more seal	southern Kefalonia	Cave KEF64
15 December 2023		southern Kefalonia	Cave KEF64
11 May 2024		northwestern Zakynthos	Cave ZAK-B

## Data Availability

None of the data were deposited in an official repository. None of the data and pictures included in the manuscript and related to a specific location of the habitat use of the Mediterranean monk seal cannot be shared, in accordance with the confidentiality related to conservation issues. Data are available upon request.

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
