# Peer review of "First Insights into the Home Range of an Adult Male Mediterranean Monk Seal Monachus monachus in the Ionian Sea, Greece, and Considerations About the Future Management of the Area"

_animals, 2025, doi:10.3390/ani15050617_

Round 1

Reviewer 1 Report

Comments and Suggestions for Authors

I found the communication very interesting. However, in my opinion, the management and conservation part mentioned in the title is missing. You reported the results but it is not clear what you would do differently for conservation. I would suggest practical measures to increase the conservation. What about the human presence and impact in the study area? I would mention that if it is something threating the monk seals. I attach the PDF with my comments.

Author Response

General answer: The title “First insights on movements of an adult male Mediterranean monk seal Monachus monachus in the Ionian Sea, Greece, and considerations about the future management of the area” was wrong and led to several misunderstandings by both reviewers. In this paper, we are examining the home range of an individual adult male monk seal and protection measures to be taken accordingly considering the fact that a special study is on-going for the stablishing an MPA in the Greek Ionian Sea. We therefore changed the title into “First insights on the home range of an adult male Mediterranean monk seal Monachus monachus in the Ionian Sea, Greece, and considerations about the future management of the area”. We also re-arranged the entire text accordingly.

Comment 1: I would expand this part in the discussion

Response 1: Lines 3-4: In the discussion, we have added several details on the future management of the area.

Comment 2:  how would you do this? Specify suggesting practical measures

Response 2: Lines 23-25: We have added recommendations on how to proceed with practical measures.

Comment 3: I would break this in two sentences maybe

Response 3: Introduction: Lines 30-34: we broke the paragraph into two sentences.

Comment 4: I would add a reference here

Response 4: Answer 4: Line 36: We have added two references.

Comment 5:  internal beach?

Response 5: Line 43: We exchanged the expression “... beach in the interior” with “internal beach”.

Comment 6: sighted?

Response 6: Line 53: we kept the word “registered” since the seal was not just sighted but photographs were taken as well along with data on the location and (partly) on day/time of the event (citizen science is not always accurate).

Comment 7: southern part of

Response 7: Line 54: we used “southern part of Kefalonia” as suggested.

Comment 8: northwestern part of

Response 8: Line 54: we used “northwestern part of Zakynthos” as suggested.

Comment 9: but in the end you don't suggest any particular conservation measure.

Response 9: Line 55: We have added several details on the future management of the area.

Comment 10:  study

Response 10: Line 57: We exchanged the word “paper” with “study” as you suggested.

Comment 11: Maybe you can add the model of the camera trap

Response 11: Line 64: We have added the various camera models we have been using up to the present time.

Comment 12: date

Response 12: Line 69: We exchanged the word “up to the present time” with “to date” as you suggested.

Comment 13: all the central Ionian Sea? And the south? Here you mention your study area as central and southern Ionian Sea

Response 13:  Line 64: Since 2018, our photo-identification project takes place in the central Ionian Sea. Zakynthos is not included here.

Comment 14: why recently?

Response 14: Line 71: This seal was identified in 2024. Photo-identification is a long-term process over generations and the monk seal is a long-lived animal. This particular seal may have reached adulthood only recently, i.e. obtaining the typical adult male monk seal characteristics so that we could distinguish it from other adult male monk seals. During the juvenile status it is very difficult to identify the seal individually.

Comment 15: I would merge all the images in one figure (all figure 2). maybe you can put first the ones in Kefalonia (figure 2a and 2b) and then the images from Zakynthos (figure 2c, 2d, 2e)

Response 15: Line 72: We re-arranged the pictures of the figures and added some more pictures from Kefalonia and from Zakynthos in order to make clear the most evident characteristics of seal A028.

Comment 16: you can remove this and just leave the table

Response 16:  Lines 73-74, Table 1: Following your suggestion, we removed the sentence.

Comment 17: maybe in the table you can merge all the sigthings of A028. Kefalonia and Zakynthos. So you have all the result in a table and it is a bit more clear

Response 17: Line 79: Following your suggestion, (a) we merged the evidence of seal A018 in both Kefalonia and Zakynthos into Table 1.

Comment 18: If you don't mention the time spent in the cave I would remove this column

Response 18: We removed the time from Table 1 as it is not relevant at all here.

Comment 19: maybe in the picture you could zoom a bit more and show the individual marks. Now in these images it is not so evident it is the same individual

Response 19: Line 83: We added Figures 4 and 5 displaying close-ups of the most evident typical characteristic of seal A028. Within the framework of this work, it is impossible to go into details for each scar, etc.

Comment 20: I would move this in the method speaking about the study area

Response 20: Lines 91-93: Following your suggestion, we moved this sentence to Material & Methods.

Comment 21: confirm that...

Response 21: Line 100: We changed this sentence according to your suggestion.

Comment 22: date

Response 22: Line 127: We exchanged the word “up to the present time” with “to date” as you suggested.

Comment 23: why? Is Zakynthos now more suitable for monk seals compared to Kefalonia? What about the human presence in the two islands?

Response 23: Lines 140-142: In Zakynthos, similar conditions prevail including human pressure. In Material & Methods, we added a paragraph about this issue. It is most probably the fact that the population is increasing, thus new habitat has to be explored and occupied.

Comment 24: I would rephrase this sentence, it is not very clear. Also what conservation measures would you suggest?

Response 24: Lines 142-145: We have added suggestions on how to proceed with practical conservation measures.

Comment 25: I would expand a bit this part in the discussion. Where will the new MPA be? What would you suggest to protect better the monk seal in the study area?

Response 25: Lines 151-134: We have added an explanation of the situation right now and about the prospects on how to proceed within the currently on-going study for establishing an MPA in the entire central and southern Ionian Sea.

Reviewer 2 Report

Comments and Suggestions for Authors

Dear Authors,

The manuscript reports on a significant event. Yes, it is certainly important. But the brevity of the presentation of the material suggests that the authors do not want to present the material they have on all the seals they recorded in their studies. It is important to explain the circumstance of the seal's movement over a distance of 10 km. In land mammals, movements are well explained in the literature. Yes, the individual was recorded in caves at remote points. But what conditions and methods were used? This should be described in detail in the manuscript. Discussion should also be supplemented for other species of pinnipeds. Long-distance movements are also known for other species. Therefore, the authors should expand the literature review. In its current form, it is insufficient. The Latin name should be added to the title of the manuscript. The authors obtained interesting results, presented them, but any fact, even a significant one, can be significantly improved. I hope my advice and recommendations will allow this manuscript to be significantly changed. I will be glad for that. After all comments have been addressed, the manuscript can be reviewed again.

Author Response

General answer: The title “First insights on movements of an adult male Mediterranean monk seal Monachus monachus in the Ionian Sea, Greece, and considerations about the future management of the area” was wrong and led to several misunderstandings by both reviewers. In this paper, we are examining the home range of an individual adult male monk seal and protection measures to be taken accordingly considering the fact that a special study is on-going for the stablishing an MPA in the Greek Ionian Sea. We therefore changed the title into “First insights on the home range of an adult male Mediterranean monk seal Monachus monachus in the Ionian Sea, Greece, and considerations about the future management of the area”. We also re-arranged the entire text accordingly.

Comment 1: Add the Latin name in brackets

Response 1: Lines 2-3: the Latin name was inserted in brackets.

Comment 2: Avoid repeating words from the manuscript title in keywords. Please replace.

Response 2: Lines 26-27: We have re-arranged the key words according to your comment.

Comment 3: First, it is necessary to introduce the reader to the topic of pinniped movements. And then move on to the object of study.

Response 3: According to our general comment (above), we did not focus on large-scale movements but on the home range of this specific monk seal. We changed the title and the text accordingly.

Comment 4: Full title (author and year) at first mention

Response 4: Line 30: The author and the year were inserted.

Comment 5: What is the approximate number?

Response 5: Line 31: The approximate (estimated) number was inserted in the first paragraph of the Introduction (line 36 of the revised manuscript).

Comment 6: It is necessary to expand this part. It is expected to offer a research perspective and ways of solving the tasks set.

Response 6: Lines 53-55:  We included here that we are dealing only with a case study on home range and in connection to protection measures to be taken within the framework of an on-going study for the protection of the central and southern Ionian Sea as a whole. We also have re-arranged the entire manuscript accordingly, further details are included in the Discussion. We do not deal in this short note with our overall results neither with long movements of the species. The short note is a case study on one single animal indeed because of the urgency to take the adequate protection measures within the on-going study for the establishment of an MPA in the central and southern Ionian Sea.

Comment 7: The physical and chemical properties of the habitat, climatic conditions, possible competitors and the diet of the seal in the habitat area should be given.

Response 7: Lines 57-59: As suggested, we inserted one paragraph about the conditions in the two locations. The physical-chemical and climatic conditions are similar in both locations and the same applies to diet and competitors.

Comment 8: It is necessary to write in detail the research methods (characteristics of the recording devices, time of day, distance from which the photographs were taken, animal behavior).

Response 8: Lines 64-67: We added details of the research method here. Animal behaviour requires specific behavioural studies and, therefore, cannot be included here. The photo-identification methodology followed is described in detail in the reference given.

Comment 9: The results are described very little. It is necessary to expand. Provide information about this individual. Write about other individuals that were recorded near this seal. What are the distances between all the studied seals except for the individual you noted. The study is based on one individual. This is very little. I understand that you have much more material? I recommend presenting it in this manuscript.

Response 9: Lines 69-92: This short note is a case study on one single animal indeed because of the urgency to take the adequate protection measures within the on-going study for the establishment of an MPA in the central and southern Ionian Sea. Further data on seal A028 in the locations described are not available (yet). The overall results within the study area of the central Ionian Sea as you have mentioned will be presented in one or more full article(s), a “short note” is much too short for this task.

Comment 10: Discussion of the results should be presented much more broadly with a review of comparisons across other pinniped species.

Response 10: Lines 100-126: The above applies here as well: this is a short note on the home range of an adult male monk seal in order to help elaborate adequate protection measures within the framework of the above-mentioned study for establishing an MPA in the central and southern Ionian Sea. The Discussion has been re-arranged according to the aims of this short note concerning the home range of seal A028 (see also above). It is due to the wrong and misleading title of our first submission that actually caused the misunderstanding in interpreting the discussion. We hope it is clear now that we only deal with the home range of this particular animal and the consequences for the elaboration of adequate protection measures.

Comment 11: Homing has long been known among land mammals [Ostfeld, Manson 1996, Andreychev, Kiyaykina 2020]. Animals return home from fairly large distances. Of course, they make long journeys to explore space, so the movements we observed in the seal can also be explained to some extent by this.

Comment 11: Homing has long been known among land mammals [Ostfeld, Manson 1996, Andreychev, Kiyaykina 2020]. Animals return home from fairly large distances. Of course, they make long journeys to explore space, so the movements we observed in the seal can also be explained to some extent by this.

Response 11: Line 137: We explained in the Discussion that, in this case, homing can be excluded.

Comment 12: This still requires more evidence. Therefore, I recommend rewriting this part less affirmatively. Yes, it is possible, but perhaps the movements of seals are much greater in distance.

Response 12: Lines 147-151: The Conclusion has been re-arranged according to the aims of this short note.

Comment 13: Add Ostfeld R.S.; Manson R.H., Long–Distance Homing in Meadow Voles, Microtus pennsylvanicus. J. Mammal. 1996, 77(3), 870-873. Andreychev, A.V.; Kiyaykina, O.S. Homing of the forest dormouse (Dryomys nitedula, Rodentia, Gliridae). Biol. Bull. Russ. Acad. Sci. 2020, 47, 1227-1234.

Response 13: We added Ostfeld & Manosn 1996 here as an example of terrestrial homing and two more references on homing in marine mammals.

Round 2

Reviewer 2 Report

Comments and Suggestions for Authors

Dear Authors,

The manuscript has been corrected and additional data has been added. It has really improved. However, the message has turned out to be quite brief. In some parts of the manuscript, the authors provide references in parentheses (surnames, year), in others in square numbers. This needs to be brought to the journal standard. The review of the research results does not cover all possible references on the topic of the research, to explain the results. But the main results of previous studies by other authors have been taken into account. Therefore, I leave these points to the editor's decision. I have no significant objections.